# EVaDE : Event-Based Variational Thompson Sampling for Model-Based Reinforcement Learning

## Abstract

Posterior Sampling for Reinforcement Learning (PSRL) is a well-known algorithm that augments model-based reinforcement learning (MBRL) algorithms with Thompson sampling. PSRL maintains posterior distributions of the environment transition dynamics and the reward function to procure posterior samples that are used to generate data for training the controller. Maintaining posterior distributions over all possible transition and reward functions for tasks with high dimensional state and action spaces is intractable. Recent works show that dropout used in conjunction with neural networks induce variational distributions that can approximate these posteriors. In this paper, we propose Event-based Variational Distributions for Exploration (EVaDE), variational distributions that are useful for MBRL, especially when the underlying domain is object-based. We leverage the general domain knowledge of object-based domains to design three types of event-based convolutional layers to direct exploration, namely the noisy event interaction layer, the noisy event weighting layer and the noisy event translation layer respectively. These layers rely on Gaussian dropouts and are inserted in between the layers of the deep neural network model to help facilitate variational Thompson sampling. We empirically show the effectiveness of EVaDE equipped Simulated Policy Learning (SimPLe) on a randomly selected suite of Atari games, where the number of agent environment interactions is limited to 100K.

## 1 Introduction

Model-Based Reinforcement Learning (MBRL) has recently gained popularity for tasks that allow for a very limited number of interactions with the environment (Kaiser et al., 2020). These algorithms use a model of the environment, that is learnt in addition to the policy, to improve sample efficiency in several ways; these include generating artificial training examples (Kaiser et al., 2020; Sutton, 1991), assisting with planning (Nagabandi et al., 2017; Coulom, 2006; Williams et al., 2015; Curi et al., 2020) and guiding policy search (Levine & Koltun, 2013; Chebotar et al., 2017). Additionally, it is easier to incorporate inductive biases derived from the domain knowledge of the task for learning the model, as the biases can be directly built into the transition and reward functions.

In this paper, we show how domain knowledge can also be used for designing exploration strategies in MBRL. Model free agents explore the space of policies and value functions; MBRL agents, on the other hand, explore the space of transition dynamics and reward functions. One method for exploring the space of transition dynamics and reward functions is Posterior Sampling for Reinforcement Learning (PSRL) (Strens, 2000; Osband & Van Roy, 2017), which uses the Thompson sampling (Thompson, 1933) method of sampling the posterior of the model to explore other plausible models. Maintaining the posterior is generally intractable and in practice, variational distributions are often used as an approximation to the posterior (Aravindan & Lee, 2021; Wang & Zhou, 2020; Zhang et al., 2019).

Traditionally, variational distributions are designed with two considerations in mind: inference and/or sampling should be efficient with the variational distribution, and the variational distribution should approximate the true posterior as accurately as possible. For MBRL, we propose to also design the

variational distribution to generate trajectories through parts of the state space that may potentially give high returns, for the purpose of exploration.

In MBRL, trajectories are generated in the state space by running policies that are optimized against the learned model. One way to generate useful exploratory trajectories, is to perturb the reward function in the model so that a different part of the state space appears to contain high rewards, causing the policy to direct the trajectories towards those states. Another method is to perturb the reward function so that parts of the state space traversed by the current policy appear sub-optimal, causing the policy to seek new trajectories.

We focus on problems where the underlying domain is object-based, i.e., domains where the reward functions depend heavily on the locations of individual objects and interactions between objects, which we call events. An example of such an object based task, is the popular Atari game Breakout. In the game, the agent is rewarded when the ball hits a brick and will not lose a life as long as the paddle successfully hits the ball, both of which can be described as an interaction between two objects. The rewards are determined by the interactions between the ball and the bricks, the wall or the paddle.

For such domains, we present **E**vent based **Va**riational **D**istributions for **E**xploration (EVaDE), a set of variational distributions that can help generate useful exploratory trajectories for deep convolutional neural network models. EVaDE comprises three Gaussian dropout (Srivastava et al., 2014) based convolutional layers, namely the noisy event interaction layer, the noisy event weighting layer and the noisy event translation layer respectively. The noisy event interaction layer is designed to provide perturbations to the reward function in states where multiple objects appear at the same location, randomly perturbing the value of interactions between objects. The noisy event weighting layer is designed to perturb the output of a convolutional layer at a single location; assuming that the output of the convolutional filters capture events, this would upweight and downweight the reward associated with these events randomly. The noisy event translation layer is designed to perturb trajectories that go through "narrow passages"; small translations can randomly perturb the returns from such trajectories, causing the policy to explore different trajectories.

These EVaDE layers can be used as normal convolutional layers and can be inserted in between the layers of the environment network models. When included in deep convolutional networks, noisy event interaction layers, noisy event weighting layers and noisy event translation layers generate perturbations on possible object interactions, on the importance of different events and on the positional importance of objects/events respectively, through the dropout mechanism which induces variational distributions over the model parameters (Srivastava et al., 2014; Gal & Ghahramani, 2016).

An interesting aspect of designing for exploration is that the variational distributions can be helpful even if they are not designed to approximate the posterior well, as long as they assist in perturbing the policy out of local optimums. After perturbing the policy, incorrect parts of the model will either be corrected by data or left incorrect if they are irrelevant to optimal behaviour.

Finally, we approximate PSRL by equipping the environment models of Simulated Policy Learning (SimPLe) (Kaiser et al., 2020) with EVaDE layers. We conduct experiments to compare EVaDE equipped SimPLe (EVaDE-SimPLe) with various popular baselines on a suite of 12 randomly selected Atari games. In the experiments conducted, all agents work in the low data regime, where the number of interactions with the real environment is restricted to 100K. EVaDE-SimPLe agents achieve a human normalized score of 0.78 on average in these games, which is 44% more than the mean score of 0.54 achieved by a recent low data regime method, CURL (Laskin et al., 2020), and 52% more than the mean score of 0.51 achieved by vanilla-SimPLe agents.

## 2 BACKGROUND AND RELATED WORK

Posterior sampling approaches like Thompson Sampling (Thompson, 1933) have been one of the more popular methods used to balance the exploration exploitation trade-off. Exact implementations of these algorithms have been shown to achieve near optimal regret bounds (Agrawal & Jia, 2017; Jaksch et al., 2010). These approaches, however, work by maintaining a posterior distribution over all possible environment models and/or action-value functions. This is generally intractable in practice. Approaches that work by maintaining an approximated posterior distribution (Osband et al., 2016b; Azizzadenesheli et al., 2018), or approaches that use bootstrap re-sampling to procure samples, (Osband et al., 2016a; Osband & Van Roy, 2015) have achieved success in recent times.

Variational inference procures samples from distributions that can be represented efficiently while also being easy to sample. These variational distributions are updated with observed data to approximate the true posterior as accurately as possible. Computationally cost effective methods such as dropouts have been known to induce variational distributions over the model parameters (Srivastava et al., 2014; Gal & Ghahramani, 2016). Consequently, variational inference approaches that approximate the posterior distributions required by Thompson sampling have gained popularity (Aravindan & Lee, 2021; Wang & Zhou, 2020; Urteaga & Wiggins, 2018; Xie et al., 2019).

Model-based reinforcement learning improves sample complexity at the computational cost of maintaining and performing posterior updates to the learnt environment models. Neural networks have been successful in modelling relatively complex and diverse tasks such as Atari games (Oh et al., 2015; Ha & Schmidhuber, 2018). Over the past few years, variational inference has been used to represent environment models, with the intention to capture environment stochasticity (Hafner et al., 2019; Babaeizadeh et al., 2018; Gregor et al., 2019).

SimPLe (Kaiser et al., 2020) is one of the first algorithms to use MBRL to train agents to play video games from images. It is also perhaps the closest to EVaDE, as it not only employs an iterative algorithm to train its agent, but also uses an additional convolutional network assisted by an autoregressive LSTM based RNN to approximate the posterior of the hidden variables in the stochastic model. Thus, similar to existing methods (Hafner et al., 2019; Babaeizadeh et al., 2018; Gregor et al., 2019), these variational distributions are used for the purpose of handling environment stochasticity rather than improving exploration. To the contrary, EVaDE-SimPLe is an approximation to PSRL, that uses a Gaussian dropout induced variational distribution over deterministic reward functions solely for the purpose of exploration. Unlike SimPLe, which uses the stochastic model to generate trajectories to train its agent, EVaDE-SimPLe agents optimize for a deterministic environment model sampled from the variational distribution. Moreover, with EVaDE, these variational distributions are carefully designed so as to explore different object interactions, importance of events and positional importance of objects/events, that we hypothesize are beneficial for learning good policies in object-based tasks.

## 3 EVENT BASED VARIATIONAL DISTRIBUTIONS FOR EXPLORATION

Event-based Variational Distributions for Exploration (EVaDE) consist of a set of variational distribution designs, each induced by a noisy convolutional layer. These convolutional layers can be inserted after any intermediate hidden layer in deep convolutional neural networks to help us construct approximate posteriors over the model parameters to produce samples from relevant parts of the model space. EVaDE convolutional layers use Gaussian multiplicative dropout to draw samples from the variational approximation of the posterior. Posterior sampling is done by multiplying each parameter, $\theta_{env}^i$, of these EVaDE layers by a perturbation drawn from a Gaussian distribution, $\mathcal{N}(1, (\sigma_{env}^i)^2)$. These perturbations are sampled by leveraging the reparameterization trick (Kingma et al., 2015; Salimans et al., 2017; Plappert et al., 2018; Fortunato et al., 2018) using a noise sample from the standard Normal distribution, $\mathcal{N}(0, 1)$, as shown in Equation 1. The variance corresponding to each parameter, $(\sigma_{env}^i)^2$, is trained jointly with the model parameters $\theta_{env}$.

$$\tilde{\theta}_{env}^i \leftarrow \theta_{env}^i(1 + \sigma_{env}^i \epsilon^i); \quad \epsilon^i \sim \mathcal{N}(0, 1) \tag{1}$$

When the number of agent-environment interactions is limited, the exploration strategy employed by the agent is critical. In object-based domains, rewards and penalties are often sparse and occur when objects interact. Hence, the agent needs to experience most of the events in order to learn a good environment model. Generating trajectories that contain events is hence a reasonable exploration strategy. Additionally, a very common issue with training using a very few number of interactions is that the agent may often get stuck in a local optima, prioritising an event, which is relatively important, but may not lead to an optimal solution. Generating potentially high return alternate trajectories that do not include that event is another reasonable exploration strategy.

With these exploration strategies in mind, we introduce three EVaDE layers, namely the noisy event interaction layer, the noisy event weighting layer and the noisy event translation layer. All the three layers are constructed with the hypothesis that the channels of the outputs of intermediate layers of deep convolutional neural networks either capture object positions, or events (interaction of multiple objects detected by multi-layer composition of the networks).

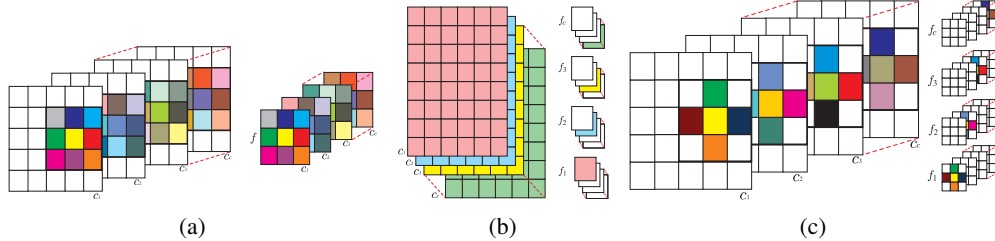

(a)                     (b)                     (c)

Figure 1: (a) This image shows one noisy event interaction filter acting on an input with $c$ channels. Here $f$ is an $m \times m$ noisy convolutional filter, which acts upon input patches at the same location across different channels, noisily altering the value of events captured at those locations.(b) This image shows how the filters of the noisy event weighting layer weight the input channels. The filters $f_1, f_2, f_3$ and $f_c$ randomly upweight and downweight the events captured by the channels $c_1, c_2, c_3$ and $c_c$ respectively. The white entries of the filter are entries that are set to zero, while the rest are trainable noisy model parameters. (c) The noisy event translation filter. The filters $f_1, f_2, f_3$ and $f_c$ noisily translate events/objects captured by the channels $c_1, c_2, c_3$ and $c_c$ respectively. The white entries of the filter are entries that are set to zero, while the rest are trainable noisy model parameters. Gaussian multiplicative dropout is applied to all the non-zero parameters of all filters.

## 3.1 NOISY EVENT INTERACTION LAYER

The noisy event interaction layer is designed with the motivation of increasing the variety of events experienced by the agent. This layer consists of noisy convolutional filters, each having a dimension of $m \times m \times c$, where $c$ is the number of input channels to the layer. Every filter parameter is multiplied by a Gaussian perturbation as shown in Equation 1. The filter dimension, $m$, is a hyperparameter that can be set so as to capture objects within a small $m \times m$ patch of an input channel. Assuming that the input channels capture the positions of different objects, a filter that combines the $c$ input channels locally captures the local object interaction within the $m \times m$ patch. By perturbing the filter, different combinations of interactions can be captured; if the filter is used as part of the reward function, it will correspondingly reward different interactions depending on the perturbation. The policy optimized for different perturbed reward functions is likely to generate trajectories that contain different events. Note that convolutional filters are equivariant, so the same filter will detect the event anywhere in the image and can result in trajectories that include the event at different positions in the image.

We describe the filter in more detail. Every output pixel of the filter, $y_{i,j}^k$, representing $(i, j)^{th}$ pixel of the $k^{th}$ output channel, can be computed as shown in Equation 2. Here $x$ is the input to the layer with $c$ input channels, $P_{x_{i,j}^l}$ is the $m \times m$ patch (represented as a matrix) centred around $x_{i,j}^l$, the $(i, j)^{th}$ pixel of the $l^{th}$ input channel, $\tilde{\theta}_k^l$ is the $l^{th}$ channel of the $k^{th}$ noisy convolutional filter, $\odot$ the Hadamard product operator, and $\mathbb{1}_m$ is an $m$ dimensional column vector whose every entry is 1.

$$y_{i,j}^k = \sum_{l=0}^{c} \mathbb{1}_m^T \left( \tilde{\theta}_k^l \odot P_{x_{i,j}^l} \right) \mathbb{1}_m \tag{2}$$

Figure 1a shows how this filter is applied over the channels of the input.

## 3.2 NOISY EVENT WEIGHTING LAYER

Overemphasis on certain events is possibly one of the main causes due to which agents converge to sub-optimal policies in object based tasks. Hence, it would be useful to easily able to increase as well as decrease the importance of an event. For this filter, we assume that each input channel is already detecting an event and design a variational distribution over model parameters that directly up-weights or down-weights the events captured by different input channels.

This layer can be implemented with the help of $c$ $1 \times 1$ noisy convolutional filters (each having a dimension of $1 \times 1 \times c$ as shown in Figure 1b), where $c$ is the number of input channels. We denote the $l^{th}$ element of the $k^{th}$ filter in the layer as $\theta_k^l$. To implement per channel noisy weighting, we set every $\theta_k^k$ as a trainable model parameter, which has a Gaussian dropout variance parameter

associated with it to facilitate noisy weighting as shown in Equation 1. All other weights, i.e., $\theta_k^l$ when $l \neq k$ are set to 0. Thus each noisy event weighting layer has $c$ trainable model parameters and $c$ trainable Gaussian dropout parameters. A pictorial representation of how this layer acts on its input is presented in Figure 1b.

Every output $y_{i,j}^k$, corresponding to the $(i,j)^{th}$ pixel of the $k^{th}$ output channel, can be computed using Equation 3, where $\tilde{\theta}_k^k$ is the noisy scaling factor for the $k^{th}$ input channel.

$$y_{i,j}^k = \tilde{\theta}_k^k x_{i,j}^k \tag{3}$$

We expect that inducing such a variational distribution that up-weights or downweights events randomly helps the agents learn from different events that are randomly emphasised by different model samples drawn from the distribution. This may eventually help them in escaping local optima caused by overemphasis of certain events.

### 3.3 NOISY EVENT TRANSLATION LAYER

In object based domains, an agent often has to perform a specific sequence of actions to successfully gain some rewards and may be penalized heavily for deviation from the sequence. We refer to the specific sequence of actions as a "narrow passage". A small translation of the positions of the environment or other objects will often cause the agent to be unsuccessful. When random translations of obstacles, events or boundaries are performed within the reward function, the optimized policy may select a different trajectory, possibly allowing it to escape from a locally optimal trajectory. We thus design the noisy event translation layer to induce a variational distribution over such model posteriors that can sample a variety of translations of relevant objects.

The noisy soft-translation on an input with $c$ channels, is performed with the help of $c$ convolutional filters, each having a dimension of $m \times m \times c$. These filters compute a noisy weighted sum of the corresponding input pixel and the pixels near it to effect a *noisy* translation of the channel. Similar to the noisy event weighting layer, each filter of the noisy event translation layer acts on one input channel. To achieve this, every parameter except the parameters of the $k^{th}$ channel of the $k^{th}$ filter, $\theta_k^k$ (which has a dimension of $m \times m$), and their corresponding dropout variances, is set to 0, for all $k$. Moreover in the channel $\theta_k^k$, only the middle column and row contain trainable parameters. Figure 1c shows a detailed pictorial representation of this structure of the filters. A random translation of up to $n$ pixels of the input can be achieved by using a $(2n+1) \times (2n+1)$ noisy event translation layer.

Equation 4 shows how $y_{i,j}^k$, the $(i,j)^{th}$ output pixel of the $k^{th}$ channel, is computed. Here, $P_{x_{i,j}^k}$ is a $m \times m$ patch centred at $(i,j)^{th}$ pixel of the $k^{th}$ input channel, $\tilde{\theta}_k^k$ is the $k^{th}$ channel of the $k^{th}$ noisy convolutional filter, $\odot$ the Hadamard product operator, and $\mathbb{1}_m$ is an $m$ dimensional column vector where all the entries are 1.

$$y_{i,j}^k = \mathbb{1}_m^T \left( \tilde{\theta}_k^k \odot P_{x_{i,j}^k} \right) \mathbb{1}_m \tag{4}$$

### 3.4 REPRESENTATIONAL CAPABILITIES OF EVADE EQUIPPED NETWORKS

Ideally, adding EVaDE layers would not cause the network to be unable to represent the true model, even if the layers are added for exploration purposes and do not accurately approximate the posterior. Theorem 1 below states that this is indeed the case.

**Theorem 1.** *Let $\mathbb{n}$ be any neural network. For any convolutional layer $l$, let $m_i(l) \times n_i(l) \times c_i(l)$ and $m_o(l) \times n_o(l) \times c_o(l)$ denote the dimensions of its input and output respectively. Then, any function that can be represented by $\mathbb{n}$ can also be represented by any network $\tilde{\mathbb{n}} \in \tilde{\mathcal{N}}$, where $\tilde{\mathcal{N}}$ is the set of all neural networks that can be constructed by adding any combination of EVaDE layers to $\mathbb{n}$, provided that, for every EVaDE layer $\tilde{l}$ added, $\tilde{l}$ uses a stride of 1, $m_i(\tilde{l}) \leq m_o(\tilde{l}), n_i(\tilde{l}) \leq n_o(\tilde{l})$ and $c_i(\tilde{l}) \leq c_o(\tilde{l})$.*

*Proof.* The proof follows from the fact that every EVaDE layer $\tilde{l}_i$ that is added is capable of representing the identity transformation. A detailed proof is presented in Appendix B. $\qquad\square$

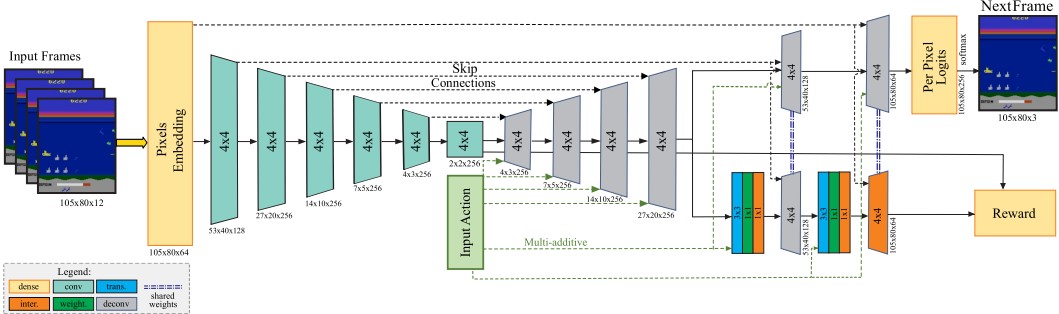

Figure 2: The network architecture of the environment model used to train EVaDE-SimPLe agents.

If the added EVaDE layers results in variational distributions that poorly approximate the posterior, performance can indeed be poorer. But with enough data, the correct model should still be learnable since it is representable, as long as the optimization does not get trapped in a poor local optimum.

### 3.5 Approximate PSRL with EVaDE equipped Simulated Policy Learning

Simulated Policy Learning (SimPLe) (Kaiser et al., 2020) is an iterative model based reinforcement learning algorithm, wherein the environment model learnt is used to generate artificial episodes to train the agent policy. In every iteration, the SimPLe agent first interacts with the real environment using its current policy. After being trained on the set of all collected interactions, the models of the transition and reward functions are then used as a substitute to the real environment to train the policy of the agent to be followed by it in its next interactions with the real environment. PSRL (Strens, 2000; Osband & Van Roy, 2017), which augments MBRL with Thompson sampling, has a very similar iterative structure as that of SimPLe. However, instead of maintaining a single environment model, PSRL maintains a posterior distribution over all possible environment models given the interactions experienced by the agent with the real environment. The agent then optimizes a policy for an environment model sampled from this posterior distribution. This policy is used in its real environment interactions of the next iteration. EVaDE equipped SimPLE approximates PSRL, by maintaining an approximate posterior distribution of the reward function with the help of the variational distributions induced by the three EVaDE layers.

Being an approximation of PSRL, EVaDE-SimPLe agent has the same iterative training structure where it interacts with the real environment using its latest policy to collect interactions, learns a transition model and an approximate posterior over the reward model parameters by jointly optimizing the environment model parameters $\theta_{env}$ and the Gaussian dropout parameters of the reward model, $\sigma_{rew}$, with the help of supervised learning and then optimizes its policy with respect to an environment characterized by the learnt transition function and a reward model sample that is procured from the posterior with the help of Gaussian dropout as shown in Equation 1. This policy is then used by the agent to procure more training interactions in the next iteration.

## 4 Experiments

We use a randomly selected suite of 12 Atari games to carry out our experiments. The test suite contains games with easy exploration such as Kangaroo, RoadRunner and Seaquest as well as BankHeist, Frostbite and Qbert, which are hard exploration games (Bellemare et al., 2016).

### 4.1 Network Architecture

In our experiments we use the network architecture of the deterministic world model introduced in Kaiser et al. (2020) to train the environment models of the SimPLe agents, but do not augment it with the convolutional inference network and the autoregressive LSTM unit. We provide a more detailed description of the network architecture in Appendix D. Readers are also referred to Kaiser et al. (2020) for more details.

The architecture of the environment model used by EVaDE agents is shown in Figure 2. This model is very similar to the one used by SimPLe agents, except that it has a combination of a $3 \times 3$ noisy event translation layer, a noisy event weighting layer and a $1 \times 1$ noisy event interaction layer inserted before the fifth and sixth de-convolutional layers. The final de-convolutional layer acts as a noisy event interaction filter when computing the reward, while it acts as a normal de-convolutional layer while predicting the next frame. Sharing weights between layers allows us to achieve this. We insert EVaDE layers in a way that it perturbs only the reward function and not the transition dynamics.

We reuse the network architecture proposed in Kaiser et al. (2020) to train the policies in both the SimPLe and EVaDE-SimPLe agents using Proximal Policy Optimization (PPO) (Schulman et al., 2017). We disable the Bernoulli dropout while training EVaDE-SimPLe agents. All the other hyperparameters used for training the policy network and environment are the same as the ones used in Kaiser et al. (2020).

## 4.2 EXPERIMENTAL DETAILS

The training regimen that we use to train all the agents is the same and is structured similarly to the the setup used by Kaiser et al. (2020). As in Kaiser et al. (2020), the agents, initialized with a random policy, collect 6400 real environment interactions before starting the first training iteration. In every subsequent iteration, every agent trains its environment model with its collection of real world interactions, refines its policy by interacting with the environment model, if it is a vanilla-SimPLe agent, or a transition model and a reward model sampled from the approximate posterior, if it is an EVaDE-SimPLe agent, and then collects more interactions with this refined policy.

PSRL regret bounds scale linearly with the length of an episode experienced by the agent in every iteration (Osband et al., 2013). Shorter horizons, however, run the risk of the agent not experiencing anything relevant before episode termination. To balance these factors, we set the total number of iterations to 30, instead of 15. We allocate an equal number of environment interactions to each iteration, resulting in 3200 agent-environment interactions per iteration. The total number of interactions that each SimPLe and EVaDE-SimPLe agent procures (about 102K) is similar to SimPLe agents trained in Kaiser et al. (2020), which allocates double the number of interactions per iteration, but trains for only 15 iterations. To disambiguate between the different SimPLe agents, we refer to the SimPLe agents trained in our paper and Kaiser et al. (2020) as SimPLe(30) and SimPLe respectively from here on.

We try to keep the training schedule of EVaDE-SimPLe and SimPLe(30) similar to the training schedule of the deterministic model in Kaiser et al. (2020) so as to keep the comparisons fair. We train the environment model for 45K steps in the first iteration and 15K steps in all subsequent iterations. Policy training is done with the help of 16 parallel PPO agents, which collect a batch of 50 environment interactions from the learnt model. In every iteration these parallel agents collect $z * 1000$ batches of interactions, where $z = 1$ in all iterations except iterations 8, 12, 23 and 27 where $z = 2$ and in iteration 30, where $z = 3$. The policy is also trained when the agent interacts with the real environment. However, the effect of these interactions on the policy (numbering 102K) is minuscule when compared to the 28.8M transitions experienced by the agent when interacting with the learnt environment model.

## 4.3 RESULTS

We train three independent runs of SimPLe(30) and EVaDE-SimPLe agents for every game in the 12 game test suite. We report the average scores achieved by these agents in Table 1 (the standard errors of these runs are given in Table 2) along with the scores achieved by the baselines SimPLe (Kaiser et al., 2020), CURL (Laskin et al., 2020), OverTrained Rainbow (Kielak, 2020) and Data-Efficient Rainbow (van Hasselt et al., 2019). For each baseline, we report the number of games in which they score more (or less) than EVaDE-SimPLe, which are counted as wins (or losses) for the baseline. We also report the mean Human Normalized Scores (HNS) achieved by all agents. Additionally, we include some visualizations that help us understand the functionality of the EVaDE layers in Appendix E.3.

EVaDE-SimPLe agents achieve the highest score in 5 of the 12 games in the test suite, outperforming every baseline in at least 8 games. Moreover, the effectiveness of the three noisy convolutional filters

Table 1: Comparison of the mean scores achieved by EVaDE-SimPLe and SimPLe(30) agents with different baselines when the number of agent-environment interactions are restricted to 100K.

| Game | SimPLe | SimPLe(30) | CURL | OTRainbow | Eff. Rainbow | EVaDE-SimPLe |
|---|---|---|---|---|---|---|
| BankHeist | 34.2 | 51.9 | 131.6 | 182.1 | 51 | **236.3** |
| BattleZone | 4031.2 | 4823 | **14870** | 4060.6 | 10124.6 | 10427 |
| Breakout | 16.4 | 19.8 | 4.9 | 9.84 | 1.9 | **25.6** |
| CrazyClimber | **62583.6** | 39591 | 12146.5 | 21327.8 | 16185.3 | 61949 |
| DemonAttack | 208.1 | 98.6 | **817.6** | 711.8 | 508 | 145.5 |
| Frostbite | 236.9 | 257.4 | **1181.3** | 231.6 | 866.8 | 262.1 |
| JamesBond | 100.5 | 255.2 | **471** | 112.3 | 301.6 | 187.5 |
| Kangaroo | 51.2 | 385 | 872.5 | 605.4 | 779.3 | **1181** |
| Krull | 2204.8 | 4460 | 4229.6 | 3277.9 | 2851.5 | **5406** |
| Qbert | 1288.8 | **3753** | 1042.4 | 509.3 | 1152.9 | 3069 |
| RoadRunner | 5640.6 | 2886 | 5661 | 2696.7 | **9600** | 8915 |
| Seaquest | 683.3 | 402.3 | 384.5 | 286.92 | 354.1 | **737.3** |
| Vs EVaDE (W/L) | 10L,2W | 10L,2W | 8L,4W | 11L,1W | 8L,4W | - |
| **HNS** | 0.36 | 0.51 | 0.54 | 0.31 | 0.38 | 0.78 |

Table 2: Scores (mean $\pm$ 1 standard error) of SimPLE agents when equipped with all three EVaDE filters individually and when equipped with all filters. All scores are averaged over three independent training runs.

| Game | SimPle (30) | Inter. Layer | Weight. Layer | Trans. Layer | EVaDE-SimPLe |
|---|---|---|---|---|---|
| BankHeist | 51.9±40.6 | 94.8±50.6 | 194.1±22.4 | 168.5 ± 26.4 | **236.3 ± 57.6** |
| BattleZone | 4823±1098 | 6375 ± 2580 | 6146±1565 | 5761 ± 705 | **10427 ± 530** |
| Breakout | 19.8±0.8 | 19.5 ± 5.5 | 19.3 ± 9.4 | 19.3 ±2.7 | **25.6 ± 4.9** |
| CrazyClimber | 39591±13350 | 59440±1441 | **71624±3115** | 63612±2980 | 61949±4594 |
| DemonAttack | 98.6±22.1 | **146.3 ±41.6** | 143.9 ± 6.2 | 99.4 ± 19.7 | 145.5 ± 22.3 |
| Frostbite | 257.4±2.8 | 246.8±4.9 | 256±3.1 | **263± 2.7** | 262.1 ± 6.3 |
| JamesBond | **255.2±8.2** | 126±81 | 161.5 ± 86.2 | 170.8 ± 104.1 | 187.5 ± 43 |
| Kangaroo | 385±301 | **1640±940** | 277.1 ± 54.8 | 1075± 529 | 1181±142 |
| Krull | 4460±826 | 3790±1200 | 4300 ±543 | 4752 ± 1727 | **5406±255** |
| Qbert | **3753±376** | 1975± 983 | 936 ±263 | 2959 ± 906 | 3069 ± 997 |
| RoadRunner | 2886±1214 | 7909±860 | 5064 ± 1268 | 8365±2150 | **8915 ± 314** |
| Seaquest | 402.3±8.6 | 590 ± 154.5 | 452.9 ± 199 | 702.7 ±76.9 | **737.3 ± 101.5** |
| **HNS** | 0.51 | 0.58 | 0.6 | 0.68 | 0.78 |

to improve exploration can be empirically verified as EVaDE-SimPLe manages to attain higher mean scores than SimPLe(30) in 10 of the 12 games, even though both methods follow the same training routine. EVaDE-SimPLe also scores a mean HNS of 0.78, which is 44% higher than the score of 0.54 achieved by the best performing baseline, CURL, and 52% more than the mean HNS of 0.51 achieved by SimPLe(30). Additionally, EVaDE-SimPLe agents also surpass the human performances (Brittain et al., 2019) in the games of CrazyClimber, RoadRunner and Krull.

## 4.4 ABLATION STUDIES

We also conduct ablation studies by equipping SimPLe(30) with just one of the three EVaDE layers to ascertain whether all of them aid in exploration. We do this by just removing the other two layers from the EVaDE environment network model (see Figure 2). Note that when the noisy event interaction filter is removed, Gaussian multiplicative dropout is not applied to the sixth de-convolutional layer.

The mean scores achieved when SimPLe(30) is equipped with only the noisy event interaction layer, the noisy event weighting layer or the noisy event translation layer individually along with the scores of SimPLe(30) and EVaDE-SimPLe are presented in Table 2.

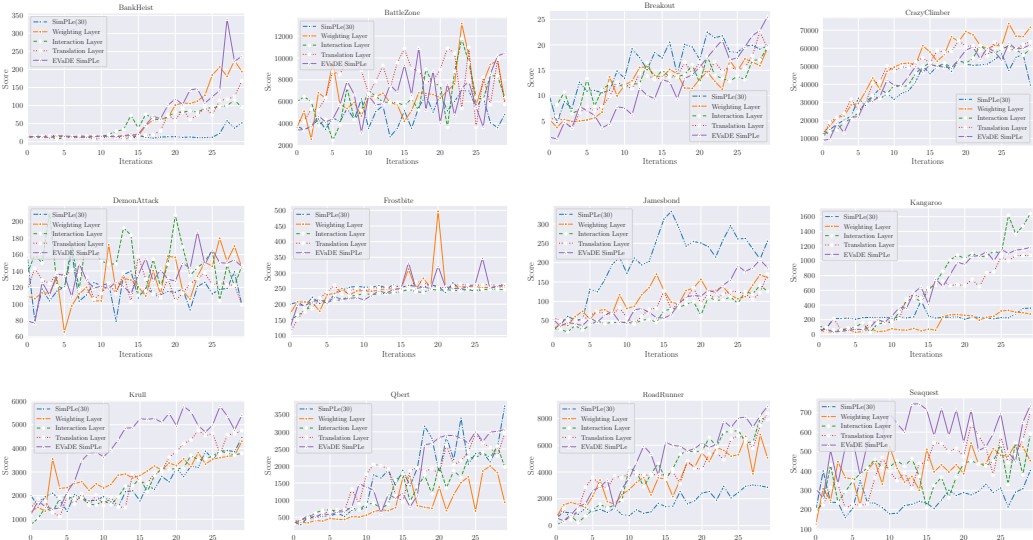

Figure 3: Learning curves of EVaDE-SimPLe agents, SimPLe(30) agents and agents which only add one of the EVaDE layers.

It can be seen that individually, each filter achieves a higher HNS than SimPLe(30), thus indicating that, on average, all the filters help in aiding exploration. Moreover, we see that with the exception of the noisy event translation layer, the increase in HNS when the other two EVaDE layers are added individually is not large. Looking at the learning curves presented in Figure 3, it can possibly be said that an increase in scores of SimPLe(30) equipped with one of the EVaDE layers at a particular iteration would mean an increase in scores of EVaDE-SimPLe, albeit in later iterations. This pattern can clearly be seen in the games of BankHeist, Frostbite, Kangaroo, Krull and Qbert. This delay in learning could possibly be attributed to the agent wasting its interaction budget exploring areas suggested by one of the layers that is ineffective for that particular game. However, we hypothesise that since all the layers provide different types of exploration, their combination is more often helpful than wasteful. This is validated by the fact that EVaDE-SimPLe achieves a higher mean HNS than any other agent in this study.

## 5    CONCLUSION

In this paper, we present Event-based Variational Distributions for Exploration (EVaDE), a set of variational distributions over reward functions. EVaDE is composed of three noisy convolutional layers, namely the noisy event interaction layer, the noisy event weighting layer and the noisy event translation layer which are designed to generate trajectories through parts of the state space that may potentially give high returns, especially in object-based domains. These layers can be inserted in between the layers of the environment network models to induce variational distributions over the model parameters that generate perturbations on object interactions, importance of events and positional importance of objects/events through the dropout mechanism. Samples drawn from these variational distributions are used to generate simulations to train the policy of a SimPLe agent.

We conduct experiments on a randomly selected test suite of 12 Atari games, where the agents are only allowed 100K interactions with the real environment. EVaDE-SimPLe agents achieve a mean human normalized score (HNS) of 0.78, which is 44 % and 52% more than the mean scores achieved by CURL and vanilla-SimPLe agents respectively. EVaDE-SimPLe agents also manage to surpass human performances in three games. We also find that each layer, when added individually to SimPLe results in a higher mean HNS. Also, the three noisy convolutional layers complement each other well, as EVaDE-SimPLe agents achieve higher mean HNS than agents which add only one noisy layer.

## REPRODUCIBILITY STATEMENT

We include the anonymized codebase that was used to conduct the experiments in the supplementary. We also mention the scores achieved by the individual runs in Table 4 in the appendix. We expect the mean scores of each EVaDE agent to fall within the range mentioned in Table 2.

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

## A  EVaDE-SimPLe as an approximation of PSRL

We present the pseudocode of EVaDE-SimPLe in Algorithm 1. As mentioned in Section 3, an EVaDE-SimPLe agent has the same iterative training structure as SimPLe and PSRL. Lines 6-10 of the algorithm show the first step of each iteration where the agent interacts with the real environment using its latest policy to collect interactions. The agent then updates its posterior distribution over the environment model parameters by jointly optimizing the environment model parameters $\theta_{env}$ which include the parameters of the transition and reward function and the Gaussian dropout parameters of the reward network $\sigma_{env}$ with the help of supervised learning (line 11). A sample from this approximate posterior distribution is then acquired with the help of Gaussian dropout as shown in lines 12-17 of the algorithm. As shown in the subsequent lines 19-26, the agent updates its policy by optimizing the parameters of the policy network, $\theta_{\pi}$ by interacting with this environment sample. This optimized policy is used by the agent to procure more training interactions by interacting with the real environment in the next iteration.

---

**Algorithm 1** Approximate PSRL with EVaDE equipped Simulated Policy Learning

---

1: Initialize agent policy, environment model and dropout parameters $\theta_{\pi}, \theta_{env}, \sigma_{env}$ respectively
2: Initialize empty real environment interaction dataset $D_{real} \leftarrow \{\}$
3: **for** iteration in $1 \cdots T$ **do**
4:     $s \leftarrow \emptyset$
5:     **while** $k_{real}$ real-world interactions not collected **do**        ▷ Interact with real environment
6:         **if** $s$ is terminal or $\emptyset$ **then**
7:             Start new episode, initialize start state $s$
8:         $a \sim \text{Policy}(s, \theta_{\pi})$
9:         $s', r \leftarrow \text{Interact\_Real\_World}(s, a)$
10:        $D_{real} \leftarrow D_{real} \cup \{(s, a, r, s')\}; s \leftarrow s'$
11:    $\theta_{env}, \sigma_{env} \leftarrow \text{Supervised\_Learn}(\theta_{env}, \sigma_{env}, D_{real})$        ▷ Learn a variational posterior
12:    **for** layer $i$ in the environment model **do**        ▷ Draw a sample from the posterior
13:        **if** $i$ is an EVaDE layer **then**
14:            Sample $\epsilon^i \sim N(0, 1)$
15:            $\tilde{\theta}^i_{env} \leftarrow \theta^i_{env}(1 + \sigma^i_{env}\epsilon^i)$
16:        **else**
17:            $\tilde{\theta}^i_{env} \leftarrow \theta^i_{env}$
18:    $s \leftarrow \emptyset, D_{sim} \leftarrow \emptyset, \text{steps} \leftarrow 0$        ▷ Train policy with environment sample
19:    **while** $k_{sim}$ interactions not completed **do**
20:        **if** $s$ is terminal or $\emptyset$ **then**
21:            Start new episode, initialize start state $s$
22:        $a \sim \text{Policy}(s, \theta_{\pi})$
23:        $s', r \leftarrow \text{Interact\_Env\_Sample}(\tilde{\theta}_{env}, s, a)$
24:        $D_{sim} \leftarrow D_{sim} \cup \{(s, a, r, s')\}, s \leftarrow s', \text{steps} \leftarrow \text{steps} + 1$
25:        **if** steps mod update\_frequency = 0 **then**
26:            $\theta_{\pi} \leftarrow \text{Reinforcement\_Learn}(\theta_{\pi}, D_{sim})$
27: **return** $\theta_{\pi}$

---

## B  Proof of Theorem 1

We provide the proof for Theorem 1, which is restated below, in this section.

**Theorem.** *Let $\mathsf{n}$ be any neural network. For any convolutional layer $l$, let $m_i(l) \times n_i(l) \times c_i(l)$ and $m_o(l) \times n_o(l) \times c_o(l)$ denote the dimensions of its input and output respectively. Then, any function that can be represented by $\mathsf{n}$ can also be represented by any network $\tilde{\mathsf{n}} \in \tilde{\mathcal{N}}$, where $\tilde{\mathcal{N}}$ is the set of all neural networks that can be constructed by adding any combination of EVaDE layers to $\mathsf{n}$, provided that, for every EVaDE layer $\tilde{l}$ added, $\tilde{l}$ uses a stride of 1, $m_i(\tilde{l}) \leq m_o(\tilde{l}), n_i(\tilde{l}) \leq n_o(\tilde{l})$ and $c_i(\tilde{l}) \leq c_o(\tilde{l})$.*

## B.1 NOTATIONS

### NEURAL NETWORKS

Any function $f$ represented by a $k$-layer neural network $\mathbb{n}$ is an ordered composition of the functions $f_1, f_2, \cdots, f_k$ computed by its layers $N_1, N_2, \cdots N_k$ respectively, i.e., $f = f_k \circ f_{k-1} \circ \cdots f_1$.

### CONVOLUTIONAL LAYERS

Any $m \times n$ convolutional layer $l$ has a total of $m \times n \times c_i(l) \times c_o(l)$ learnable parameters, where $c_i(l)$ and $c_o(l)$ are the number of channels in the input and output of the layer respectively. The parameters of any convolutional layer $l$ can be partitioned into $c_o(l)$ filters, where each filter has $m \times n \times c_i(l)$ parameters, and is responsible for computing one output channel.

We denote the set of parameters of any convolutional layer by $\theta$. We denote the set of parameters of the $k^{th}$ filter by $\theta_k$, and the parameters of the $l^{th}$ channel of this filter by $\theta_k^l$. We denote the $(i, j)^{th}$ parameter of the $l^{th}$ channel of the $k^{th}$ filter by $\theta_k^{l,i,j}$. For noisy convolutional layers, we have a learnable Gaussian dropout parameter attached to every parameter of the convolutional layer (see Equation 1). We use $\sigma_k$, $\sigma_k^l$ and $\sigma_k^{l,i,j}$ to denote the dropout parameters of the $k^{th}$ filter, the $l^{th}$ channel of the $k^{th}$ filter and the $(i, j)^{th}$ parameter of the $l^{th}$ channel of the $k^{th}$ filter respectively.

### STRIDES

A stride is a hyperparameter of a convolutional layer, that determines the number of pixels of the input that each convolutional filter moves, to compute the next output pixel.

## B.2 IMPLICATIONS OF THE CONSTRAINTS IN THEOREM 1

Theorem 1 states that every EVaDE layer $\tilde{l}$ added uses a stride of 1 and satisfies the constraints $m_i(\tilde{l}) \leq m_o(\tilde{l}), n_i(\tilde{l}) \leq n_o(\tilde{l})$ and $c_i(\tilde{l}) \leq c_o(\tilde{l})$. This means that for any inserted EVaDE layer, every output dimension is at least as large as its corresponding input dimension. This eventually implies that for every EVaDE layer, all input and output dimensions match, i.e., $m_i(\tilde{l}) = m_o(\tilde{l}), n_i(\tilde{l}) = n_o(\tilde{l})$ and $c_i(\tilde{l}) = c_o(\tilde{l})$.

To see why, let us assume that the EVaDE layers $\tilde{l}_j, \cdots \tilde{l}_k$ are inserted, in order, in between the layers $N_i$ and $N_{i+1}$ of a neural network $\mathbb{n}$. As $N_i$ and $N_{i+1}$ are two consecutive layers of $\mathbb{n}$, we must have $m_i(N_{i+1}) = m_o(N_i), n_i(N_{i+1}) = n_o(N_i)$ and $c_i(N_{i+1}) = c_o(N_i)$. This implies that the dimensions of the input to layer $\tilde{l}_j$ match the dimensions of the output of the layer $\tilde{l}_k$, i.e., $m_i(\tilde{l}_j) = m_o(\tilde{l}_k), n_i(\tilde{l}_j) = n_o(\tilde{l}_k)$ and $c_i(\tilde{l}_j) = c_o(\tilde{l}_k)$. However, under the constraints imposed in Theorem 1, every output dimension is greater than or equal to its corresponding input dimension for every EVaDE layer. Thus, matching the output dimensions of $\tilde{l}_k$ with the input dimensions of $\tilde{l}_j$ is only possible if the input and output dimensions match for every EVaDE layer $\tilde{l}_j, \cdots \tilde{l}_k$ that is added.

With the above implications, the constraint of using a stride of 1, forces SAME padding for every EVaDE layer, and also ensures that patches centred around every $(i, j)^{th}$ pixel of every channel in the input are used to compute the outputs. This is an important implication that will help us prove the claims that all EVaDE layers can represent the identity transformation.

## B.3 CLAIMS

We prove the three following claims by construction, i.e., showing that there is a combination of parameters using which these layers can perform the identity transformation.

**Claim 1.** *The noisy event interaction layer can represent the identity transformation.*

*Proof.* Let us assume an $m \times m$ noisy event interaction layer. With the help of the observations made in the previous section, we are ensured of using patches centred around every input $x_{i,j}^l \ \forall i, j, l$ and the constraints also ensure that the number of filters in this layer is equal to the number of input channels.

The identity transformation can be achieved with the following parameter assignments.

- The dropout parameter $\sigma_k^{l,i,j}$ corresponding to every convolutional layer parameter $\theta_k^{l,i,j}$ is set to zero.

- The layer parameter corresponding to the central entry of the $k^{th}$ layer of the $k^{th}$ convolutional filter, i.e., $\theta_k^{k,\lceil \frac{m}{2} \rceil, \lceil \frac{m}{2} \rceil}$ is set to 1 $\forall k$.

- All other convolutional layer parameters are set to 0.

As stated in Equation 2, the event interaction layer computes the outputs $y_{i,j}^k$ $\forall i, j, k$ using the following equation.

$$y_{i,j}^k = \sum_{l=0}^c \mathbb{1}_m^T \left( \tilde{\theta}_k^l \odot P_{x_{i,j}^l} \right) \mathbb{1}_m$$

Applying the above parameter assignments, we get $y_{i,j}^k = x_{i,j}^k$, as the only non-zero parameter in the $k^{th}$ filter, that is set to 1, aligns with $x_{i,j}^k$. This is the required identity transformation.

$\square$

**Claim 2.** *The noisy event weighting layer can represent the identity transformation.*

*Proof.* The noisy event weighting layer uses $c$ $1 \times 1$ convolutional filters, where $c$ is the number of input channels. Consequently, $\theta_k^k$, is just a single trainable parameter instead of a grid of trainable parameters as in the other two EVaDE layers.

The identity transformation can be achieved with the following parameter assignments.

- The dropout parameter $\sigma_k^l$ corresponding to every convolutional layer parameter $\theta_k^l$ is set to zero.

- The layer parameter corresponding to the $k^{th}$ layer of the $k^{th}$ convolutional filter, i.e., $\theta_k^k$ is set to 1 $\forall k$.

- All other convolutional layer parameters are set to 0.

This is a valid assignment, as the only parameters set to 1 are trainable, while the other parameters are forced to be set to 0 by construction (see Section 3.2).

As stated in Equation 3, the event interaction layer computes every output $y_{i,j}^k$ using the following equation.

$$y_{i,j}^k = \tilde{\theta}_k^k x_{i,j}^k$$

Setting $\theta_k^k = 1$ and $\sigma_k^k = 0$ $\forall k$, yields $y_{i,j}^k = x_{i,j}^k$ $\forall i, j, k$ , which is the identity transformation required. $\square$

**Claim 3.** *The noisy event translation layer can represent the identity transformation.*

*Proof.* In this case, we can use the parameter assignments as stated in the proof of Claim 1 to produce an identity transformation. This is possible, since we construct the noisy event translation layer with the same structure of an $m \times m$ convolutional layer with the number of filters equalling the number of input channels. Moreover, the only non-zero parameter (which is set to 1) in the $k^{th}$ filter, $\theta_k^{k,\lceil \frac{m}{2} \rceil, \lceil \frac{m}{2} \rceil}$ is in the middle row and middle column of its $k^{th}$ channel, making it a valid assignment for the noisy event translation layer (see Section 3.3).

As stated in Equation 4, the event interaction layer computes every output $y_{i,j}^k$ using the following equation.

$$y_{i,j}^k = \mathbb{1}_m^T \left( \tilde{\theta}_k^k \odot P_{x_{i,j}^k} \right) \mathbb{1}_m$$

As in the case with the noisy event interaction layer, substituting these assignments, we get $y_{i,j}^k = x_{i,j}^k \ \forall i, j, k$ , which is the identity transformation required.

$\square$

### B.3.1 PROOF OF THEOREM 1

We have to prove that all elements from $\tilde{\mathcal{N}}$, the set of neural networks that can be constructed by adding any combination of EVaDE layers to the neural network n, can represent the functions represented by k-layered neural network n.

Let $\tilde{n}$ be a general element from $\tilde{\mathcal{N}}$, that adds the EVaDE layers $\tilde{l}_1, \tilde{l}_2, \cdots \tilde{l}_m$, in order, after the layers $N_{i_1}, N_{i_2}, \cdots N_{i_m}$ of the neural network n, where $i_{j-1} \leq i_j \leq i_{j+1}$ ; $\forall 2 \leq j \leq m-1$ and $i_1 \geq 0, i_m \leq k$. Adding an EVaDE layer after $N_0$ refers to it being added after the input layer and before the first layer of n. Note that more than one EVaDE layer can be added after any layer $N_j$ of n.

Also, let $f_1, f_2, \cdots f_k$ be the functions computed by the layers $N_1, N_2, \cdots, N_k$ of n respectively. Thus the function represented by n is $f = f_k \circ f_{k-1} \circ \cdots f_1$.

Let $\tilde{f}_1, \tilde{f}_2, \cdots \tilde{f}_m$ be the functions computed by the EVaDE layers $\tilde{l}_1, \tilde{l}_2, \cdots \tilde{l}_m$ respectively. Thus the function computed by $\tilde{n}$ is $\tilde{f} = f_k \circ f_{k-1} \circ \cdots \circ \tilde{f}_m \circ f_{i_m} \cdots \circ \tilde{f}_1 \circ f_{i_1} \circ \cdots f_1$. As all $\tilde{f}_1, \tilde{f}_2, \cdots \tilde{f}_m$ can learn to represent the identity transformation, $\tilde{f}$ can learn to represent $f$. This implies that $\tilde{n}$ can represent any function represented by n.

## C VARIATIONAL DISTRIBUTIONS USING DROPOUTS

Variational methods are used to approximate inference and/or sampling when using intractable posterior distributions. These methods work by using variational distributions that facilitate easy sampling and/or inference, while approximating the true posterior as closely as possible.

These methods require the user to define two distributions, the prior $p(\theta)$, and the variational distribution $q(\theta)$. Given a set of training samples $D = (X, Y)$, where $X$ is the set of input samples and $Y$ the set of corresponding labels, variational methods work to minimize the KL-divergence between the learnt variational distribution $q(\theta)$ and the true posterior $p(\theta|D)$. This is equivalent to maximizing the Evidence Lower Bound (ELBO) as shown below.

$$KL(q(\theta), p(\theta|D)) = \int q(\theta) \log \frac{q(\theta)}{p(\theta|D)} d\theta$$

Now,

$$p(\theta|D) = p(\theta|X, Y) = \frac{p(\theta)p(X, Y|\theta)}{p(X, Y)} = \frac{p(\theta)p(Y|X, \theta)p(X|\theta)}{p(X, Y)}$$
$$= \frac{p(\theta)p(Y|X, \theta)p(X)}{p(X, Y)};$$

Substituting the value for $p(\theta|D)$,

$$KL(q(\theta), p(\theta|D)) = \int q(\theta) \left[ \log \frac{q(\theta)p(X, Y)}{p(\theta)p(Y|X, \theta)p(X)} \right] d\theta$$
$$= \int q(\theta) \log \frac{q(\theta)}{p(\theta)p(Y|X, \theta)} d\theta + \int q(\theta) \log \frac{P(X, Y)}{P(X)} d\theta$$
$$= \int q(\theta) \log \frac{q(\theta)}{p(\theta)p(Y|X, \theta)} d\theta + \log \frac{P(X, Y)}{P(X)}$$
$$= \int q(\theta) \log \frac{q(\theta)}{p(\theta)} d\theta - \int q(\theta) \log p(Y|X, \theta) d\theta + \log \frac{P(X, Y)}{P(X)}$$
$$= KL(q(\theta), p(\theta)) - \int q(\theta) \log p(Y|X, \theta) d\theta + \log \frac{P(X, Y)}{P(X)}$$

Since $P(X, Y)$ and $P(X)$ are constants with respect to $\theta$, the set of parameters that minimize $KL(q(\theta), p(\theta|D))$ are the same as the ones that maximize the ELBO, i.e.,

$$\arg\min_{\theta} KL(q(\theta), p(\theta|D)) = \arg\max_{\theta} \int q(\theta) \log p(Y|X, \theta)d\theta - KL(q(\theta), p(\theta))$$

### C.1 Dropouts as Variational Distributions

Gal & Ghahramani (2016) introduces the usage of dropout as a mechanism to induce variational distributions, samples from which are used to approximate the ELBO. The first term of the ELBO can be re-written as,

$$\int q(\theta) \sum_{i=1}^{N} \log p(y_i|x_i, \theta)d\theta$$

where every $(x_i, y_i)$ is a training example from $D$.

This integral can be approximated by averaging out the log-probabilities using several samples drawn from the variational distribution $q(\theta)$ (Equation 5).

$$\int q(\theta) \sum_{i=1}^{N} \log p(y_i|x_i, \theta)d\theta \approx \sum_{i=1}^{N} \log p(y_i|x_i, \theta_i); \text{ where } \theta_i \sim q(\theta) \tag{5}$$

Neural networks that use different types of dropouts help us maintain variational distributions $q(\theta)$ that approximate posteriors over deep Gaussian processes (Gal & Ghahramani, 2016; Kingma et al., 2015). Procuring a sample from this posterior using $q(\theta)$ is easy, as a random dropped out network corresponds to a sample from the posterior over these deep Gaussian processes.

## D  Network Architectures

In this section, we detail the network architectures used for training the environment models of SimPLe (Kaiser et al., 2020) and EVaDe-SimPLe, and the policy network architectures used by both the methods.

### D.1  Environment Network Architecture

#### D.1.1  SimPLe

In our experiments we use the network architecture of the deterministic world model introduced in Kaiser et al. (2020) to train the environment models of the SimPLe agents, but do not augment it with the convolutional inference network and the autoregressive LSTM unit.

Given four consecutive game frames and an action as input, the network jointly models the transition and reward functions, as it predicts the next game frame and the reward using the same network. The network consists of a dense layer, which outputs a pixel embedding of the stacked input frames. This layer is followed by a stack of six $4 \times 4$ convolutional layers, each with a stride of 2. These layers are followed by six $4 \times 4$ de-convolutional layers. For $1 \leq i \leq 5$, the $i^{th}$ de-convolutional layers, take in as input, the output of the previous layer, as well as the output of the $6 - i^{th}$ convolutional layer. The last de-convolutional layer takes in as input the output of its previous layer and the dense pixel embedding layer. An embedding of the action input is multiplied and added to the input channels of every de-convolutional layer. The outputs from the last de-convolutional layer is passed through a softmax function to predict the next frame. The outputs from the last de-convolutional layer is also combined with the output of the last convolutional layer and then passed through a fully connected layer with 128 units followed by the output layer to predict the reward.

#### D.1.2  EVaDE-SimPLe

The architecture of the environment model used by EVaDE agents is shown in Figure 2. This model resembles the model of SimPLe agents until the fourth de-convolutional layer. All the stand-alone

EVaDE layers that we use, use a stride of 1 and SAME padding so as to keep the size of the inputs and outputs of the layer same. As the EVaDE layers are added only to the reward function, we split the network into two parts, one that predicts the next frame (the transition network) and one that predicts the reward (the reward network) respectively. We denote the last two de-convolutional layers in each part $d_5^t$, $d_6^t$ and $d_5^r$, $d_6^r$ respectively.

As shown in Figure 2, in the transition network, the outputs of the fourth de-convolutional layer and the first convolutional layer are passed to $d_5^t$. $d_6^t$ takes in as inputs the outputs of $d_5^t$ and the pixel embedding layer.

The reward network adds a combination of a $3 \times 3$ noisy event translation layer, a noisy event weighting layer and a $1 \times 1$ noisy event interaction layer which are inserted before both $d_5^r$ and $d_6^r$. $d_5^r$ shares weights with $d_5^t$, and takes in the outputs of the previous event interaction layer and the first convolutional layer as inputs. Likewise, $d_6^r$ shares weights with $d_6^t$, and takes in the outputs of the previous event interaction layer and the pixel embedding layer as inputs. Moreover, we also apply Gaussian multiplicative dropout to the weights of $d_6^r$, to make it act as an event interaction layer. As with SimPLe agents, an embedding of the action input is multiplied and added to the input channels of every de-convolutional layer (also shown in Figure 2).

The outputs of $d_6^t$ are passed through a softmax to predict the next frame, while the outputs of $d_6^r$ are combined with the output of the last convolutional layer and passed through a fully connected layer with 128 units followed by the output layer to predict the reward.

### D.2 POLICY NETWORK

The policy network for both SimPLe and EvADE-SimPLe agents consists of two convolutional layers followed by a hidden layer and an output layer. The inputs to the policy network are four consecutive game frames, which are stacked and passed through two $5 \times 5$ convolutional layers, both of which use a stride of 2. These convolutional layers are followed by a fully connected layer with 128 hidden units, which is followed by the output layer, that predicts the stochastic policy, i.e., the probabilities corresponding to each valid action, and the value of the current state of the agent.

## E EXPERIMENTAL DETAILS

### E.1 HUMAN NORMALIZED SCORE

We use the human normalized scores from Brittain et al. (2019) as defined in Equation 6 to compare our agents.

$$\text{HNS}_{\text{agent}} = \frac{\text{Score}_{\text{agent}} - \text{Score}_{\text{random}}}{\text{Score}_{\text{human}} - \text{Score}_{\text{random}}} \qquad (6)$$

where $\text{Score}_{\text{agent}}$, $\text{Score}_{\text{human}}$ and $\text{Score}_{\text{random}}$ denote the scores achieved by agent being evaluated, a human and an agent which acts with a random policy respectively.

We also list the baseline scores achieved by humans and random agents, as listed in Brittain et al. (2019) in Table 3 for easy access.

### E.2 MORE EXPERIMENTAL DETAILS

We present the scores achieved by all three independent runs of all agents trained in Table 4 . Figure 4 shows the learning curves as shown in Figure 3 with error bars equal to a width of 1 standard error on each side.

Table 3: Baseline human and random values used to calculate Human Normalized Scores

| Game | Human Score | Random Score |
|---|---|---|
| BankHeist | 753.1 | 14.2 |
| BattleZone | 37187.5 | 2360 |
| Breakout | 30.5 | 1.7 |
| CrazyClimber | 35829.4 | 10780.5 |
| DemonAttack | 1971 | 152.1 |
| Frostbite | 4334.7 | 65.2 |
| JamesBond | 302.8 | 29 |
| Kangaroo | 3035 | 52 |
| Krull | 2665.5 | 1598 |
| Qbert | 13455 | 163.9 |
| RoadRunner | 7845 | 11.5 |
| Seaquest | 42054.7 | 68.4 |

Table 4: Scores achieved by every independent run of every SimPLE agent and when equipped with different EVaDE layers

| Game | SimPLe(30) | Inter. Layer | Weight. Layer | Trans. Layer | EVaDE-SimPLe |
|---|---|---|---|---|---|
| BankHeist | 133.1 | 85 | 232.2 | 218.4 | 155.3 |
| | 9.375 | 12.5 | 195.3 | 128.8 | 205.9 |
| | 13.13 | 186.9 | 154.7 | 158.4 | 347.8 |
| BattleZone | 4156 | 1313 | 9250 | 4438 | 10844 |
| | 6969 | 8031 | 4250 | 6000 | 9375 |
| | 3344 | 9781 | 4938 | 6844 | 11063 |
| Breakout | 20.09 | 8.563 | 29.78 | 14.45 | 35.38 |
| | 18.25 | 25.03 | 27.56 | 23.64 | 20.91 |
| | 20.94 | 24.81 | 0.625 | 19.69 | 20.5 |
| CrazyClimber | 54569 | 57534 | 75300 | 69494 | 55194 |
| | 51244 | 58522 | 74141 | 59838 | 59934 |
| | 12959 | 62266 | 65431 | 61503 | 70719 |
| DemonAttack | 55.31 | 134.1 | 215.3 | 129.7 | 169.1 |
| | 112.7 | 155.5 | 71.41 | 105.9 | 100.9 |
| | 127.7 | 142.2 | 152.2 | 62.5 | 166.4 |
| Frostbite | 261.3 | 256.6 | 250 | 263.4 | 268.4 |
| | 251.9 | 241.6 | 259.4 | 258.1 | 249.4 |
| | 259.1 | 242.2 | 259.1 | 267.5 | 268.4 |
| JamesBond | 268.8 | 12.5 | 59.38 | 371.9 | 232.8 |
| | 240.6 | 282.8 | 332.8 | 117.2 | 101.6 |
| | 256.3 | 82.81 | 92.19 | 23.44 | 228.1 |
| Kangaroo | 987.5 | 3294 | 293.8 | 25 | 1144 |
| | 56.25 | 1588 | 362.5 | 1481 | 956.3 |
| | 112.5 | 37.5 | 175 | 1719 | 1444 |
| Krull | 5639 | 6124 | 5150 | 5548 | 5569 |
| | 4873 | 3103 | 3290 | 7266 | 4906 |
| | 2868 | 2142 | 4460 | 1443 | 5744 |
| Qbert | 3002 | 3935 | 516.4 | 1193 | 1082 |
| | 4151 | 1133 | 1430 | 4190 | 3916 |
| | 4106 | 857 | 873.4 | 3494 | 4208 |
| RoadRunner | 2793 | 8794 | 3034 | 5709 | 8666 |
| | 831.3 | 8744 | 4763 | 6763 | 8541 |
| | 5034 | 6188 | 7397 | 12622 | 9538 |
| Seaquest | 392.5 | 791.3 | 221.3 | 649.4 | 536.3 |
| | 419.4 | 286.3 | 288.1 | 604.4 | 813.8 |
| | 395 | 692.5 | 849.4 | 854.4 | 861.9 |

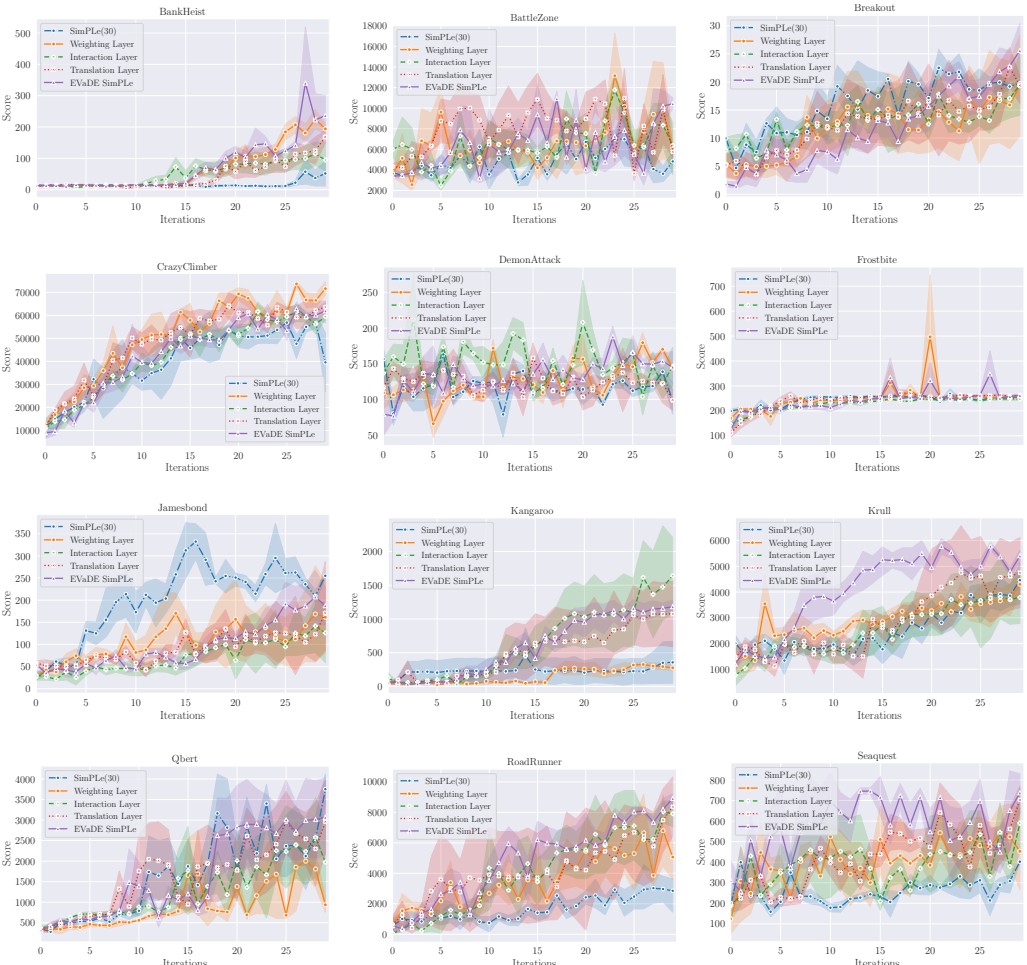

Figure 4: Learning curves of EVaDE-SimPLe agents, SimPLe(30) agents and agents which only add one of the EVaDE layers with error bars of 1 standard error.

### E.3 VISUALIZATIONS OF THE EVaDE LAYERS

We present some visualizations of the input and output feature maps of the noisy event interaction layer, the noisy event weighting layer and the noisy event translation layer. All these visualizations were obtained from the final trained model, with the learned weights and variances.

In Figures 5 and 6 we show illustrations of an output feature map detecting interactions between the right facing green-coloured enemy ships and the right facing blue-coloured divers given different input images from the game of Seaquest. We also show two input feature maps, which seem to capture the positions of these objects at the same locations. We observe that the pixels in the output feature map in Figure 5 are brighter at the locations where the two objects are close to each other, whereas in the same feature map these pixels are dimmer when the two objects are separated by some distance (in Figure 6).

In Figures 7 and 8, we show two feature maps before and after passing them through a noisy event weighting layer. The inputs for these visualizations were taken from the game of Breakout. The input feature maps to the noisy event weighting layer seem to capture the bricks from the input image. The output feature map in Figure 7 is an upweighted version of its input, as the pixels seem to be brighter. On the other hand, the output feature map in Figure 8 seems to down-weight its input feature map, as the pixels seem a lot dimmer. The weighting factors for the input-output pairs shown in Figures 7 and 8 are 1.93 and 0.57 respectively.

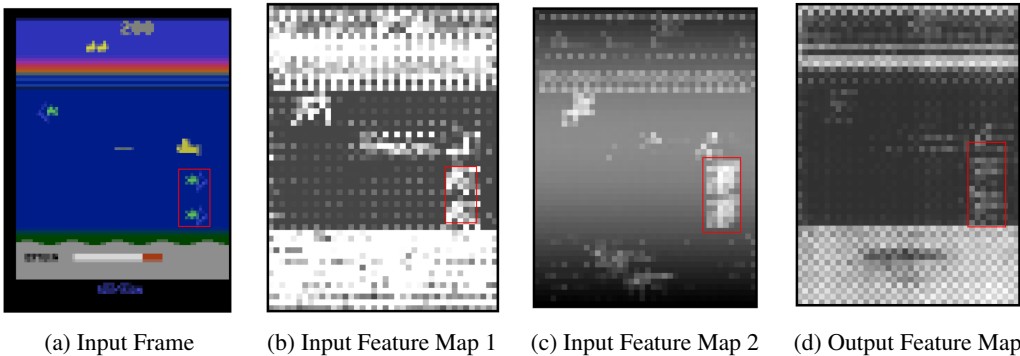

| (a) Input Frame | (b) Input Feature Map 1 | (c) Input Feature Map 2 | (d) Output Feature Map |

Figure 5: This figure shows an output feature map(channel) that captures interactions between two input feature maps when passed through the noisy event weighting layer. Here, the interaction between the blue *diver* and the green *enemy* is captured in (d).

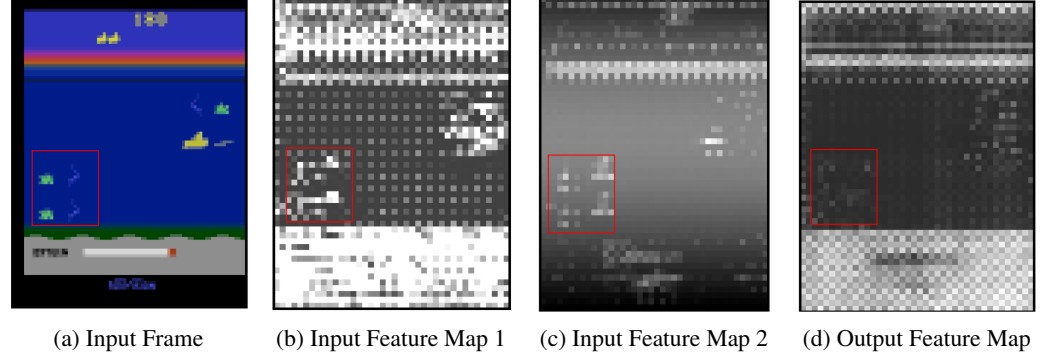

| (a) Input Frame | (b) Input Feature Map 1 | (c) Input Feature Map 2 | (d) Output Feature Map |

Figure 6: This figure shows the same output feature map of the noisy event interaction layer as shown in Figure 5, but when there is no interaction between the blue and green objects. The blue and green objects in (a) are separated by some distance, and thus the interaction is not shown in (d).

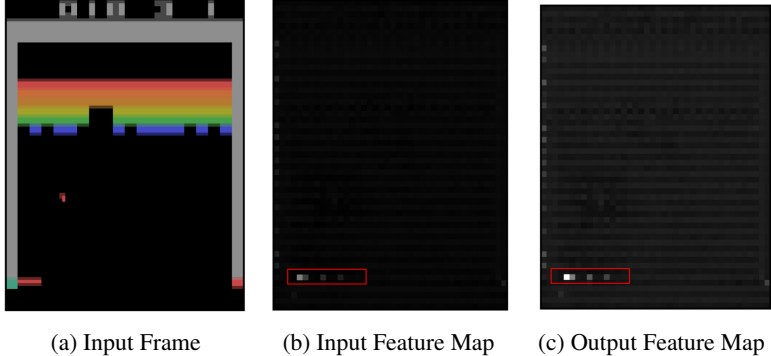

| (a) Input Frame | (b) Input Feature Map | (c) Output Feature Map |

Figure 7: This figure shows an output feature map (channel) that up-weights the corresponding input feature map when passed through the noisy event weighting layer.

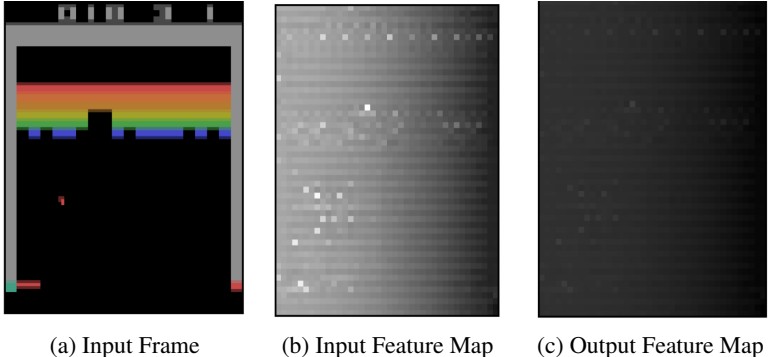

(a) Input Frame      (b) Input Feature Map      (c) Output Feature Map

Figure 8: This figure shows an output feature map (channel) that down-weights the corresponding input feature map when passed through the noisy event weighting layer.

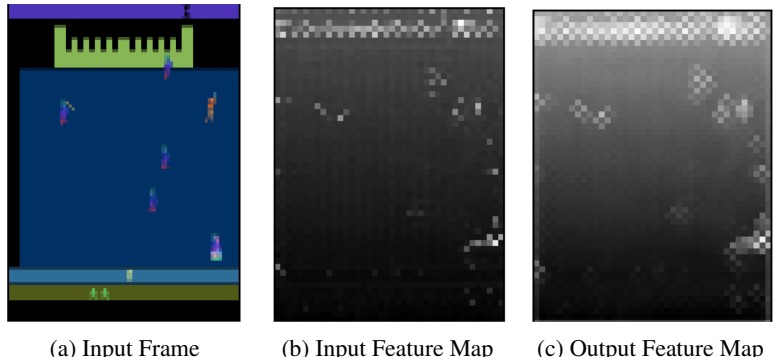

(a) Input Frame      (b) Input Feature Map      (c) Output Feature Map

Figure 9: This figure shows the function of the noisy translation layer. The output feature map translates the input pixels to its top, bottom, left and right to different degrees

In Figure 9, we show the input and output feature maps of a game state from Krull, before and after passing it through a noisy event translation layer. The input feature map seems to capture different objects from the input image. The translation effect in output feature map can be seen clearly as every light pixel in the input seems to have lightened up the pixels to its top, bottom, left and right to different degrees.

### E.4 CODEBASE USED AND HYPERPARAMETERS

We build our SimPLe and EVaDE-SimPLe agents by utilizing the implementation of SimPLe agents from Vaswani et al. (2018). To keep the comparison fair, we use the same hyperparameters as used by Vaswani et al. (2018) to train all our agents. The codebase in Vaswani et al. (2018) uses an Apache 2.0 license, thus allowing for public use and extension of their codebase.

### E.5 COMPUTATIONAL HARDWARE USED

We train our agents on a cluster of 4 NVIDIA RTX 2080 Ti GPUs with an Intel Xeon Gold 6240 CPU. The total time taken to train 3 independent runs of all 5 algorithms on the test suite of 12 games was around 79 days and 20 hours.

