# OpenReview forum: "EVaDE : Event-Based Variational Thompson Sampling for Model-Based Reinforcement Learning"
_ICLR.cc/2022/Conference — ICLR 2022 Submitted_

### Official Review · Reviewer_XiQT · 2021-10-28

**Correctness:** 3
**Technical Novelty And Significance:** 2
**Empirical Novelty And Significance:** 3
**Recommendation:** 6
**Confidence:** 3

**Main Review:**

Strengths:
- Using dropout to approximate posteriors and apply Thompson sampling to MBRL. (I am just curious to know whether it has already been applied in this context, see question 1. below)
- I also like that only 3 additional layers are needed and can be added to any model.
- the paper is well written and easy to read.
- The experimental setup is well detailled (see specific comments below)
- the different ablation studies and in particular the ablation study of table 2 showing that using the 3 layers is good in general. Even though sometimes only one of the layers leads to better performance, using the 3 layers improves the performance of Simple(30) on most of the games.

Weaknesses:
- the technique seems to be only applicable to object-based tasks. However I like that it incorporates domain knowledge.

Minor comments:
- Please put the object-based task example in the introduction. I had to wait until section 3 to understand what the author were meaning by this.
- Table 1 title: mean scores of achieved -> mean scores achieved

Questions:
1. Are the authors aware of any other work applying dropout to perform Thompson sampling in MBRL?
2. In algorithm 1 it seems that the same posterior sample is used for all the episodes when training the policy with the environment. Isn't there a risk of the policy overfitting to this specific model? For instance the idea of Model ensemble TRPO (https://arxiv.org/abs/1802.10592) was to pick one member of the ensemble at random at each step. I understand that we would not want to do this at each step but we could imagine having a different posterior sample for each different episode. On the other hand, this could seem to be contrary to the idea of Thompson sampling where we would like the policy to use one posterior sample to drive its exploration.
3. Is there a reason explaining that Evade-Simple underperforms Simple(30) in 2 games?
4. Why approximating a posterior for the reward function part only? Wouldn't it be also possible to use it for the dynamics?

Section 4.2
- the agents collect 6400 real environment interactions: please add that this is with the randomly initialized policy
- paragraph 2: I am not sure to understand why the horizon length tradeoff influences the number of iterations.
- "we train the environment model for 45K steps in the first iteration and 15K steps in all subsequent iterations": does the authors mean that they use the past 45K steps to train the environment model? I am not sure, only 6400 interactions are collected before the first training iteration.
- why is "z" changing?

**Summary Of The Paper:**

This paper proposes an approach for Model-based reinforcement learning relying on Posterior Sampling for Reinforcement learning. Posterior sampling for reinforcement learning uses Thompson sampling to balance exploitation and exploration. However this requires maintaining posterior of dynamic transitions which is often intractable. The authors use dropout, which has been shown to approximate posteriors, to maintain approximate posteriors of the transitions and the reward function. The system model presented in the paper specifically applies to object-based tasks. Only three event convolutional layers use dropout: one for object interaction, one for event weighting and one for event translation. These layers can be inserted into any existing neural network model to provide an approximate posterior. The authors combine this model with a PPO agent trained on the model to show its empirical performance on Atari games with a limited number of interactions with the real environment.

**Summary Of The Review:**

I liked the paper. My grade is explained by the fact that I am not very familiar with the domain of vision and games, hence I might be unaware of important related works. I'll upgrade my score from the discussion with the other reviewers and the authors.

---

> ### Author Response · Authors · 2021-11-14
> **Response to Reviewer XiQT (Part 1 of 2)**
>
> We thank the reviewer for the detailed and informative review and pertinent suggestions. We address the questions raised below.
>
> - Are the authors aware of any other work applying dropout to perform Thompson sampling in MBRL?\
> We are aware of dropouts being used to approximate Thompson sampling in model-free RL [1]. However, to the best of our knowledge,  this is the first work that has successfully used dropout induced variational distributions for Thompson sampling in  MBRL methods.
>
> &nbsp;
> - In algorithm 1, why was only one sample from the posterior used to train the policy with the environment model?\
> EVaDE-SimPLe is an approximation of PSRL. PSRL follows an iterative process, where each iteration has 4 stages, (1) It samples a model from the current posterior distribution, (2) learns an optimal policy for that sampled model, (3) uses the learnt policy for subsequent interactions with the real environment to collect more data and (4) updates posterior with collected data.
> While training the policy with the environment model in EVaDE-SimPle, we wish to learn an optimal policy for the sampled model, as mentioned in step (2) above, and thus use one posterior sample for this phase. This optimized policy will be used for data collection when the agent interacts with the real environment subsequently.
> We are aware of the ensemble or particle-based approximations of Thompson sampling that maintain different sets of parameters for each particle of ensemble member [1,2]. Usually, these methods try to approximate the posterior with the particles as point estimates, each of which could possibly occupy different parts of the parameter space.
>
> &nbsp;
> - Is there a reason explaining that Evade-Simple underperforms Simple(30) in 2 games?\
> It is possible that some layers are less effective in some games, leading to the agent wasting its interaction budget exploring areas suggested by one of the less effective layers for that particular game. As such, in the two games, it is possible that the layers explore excessively leading to lower scores. However, on average, using all three EVaDE layers seems more beneficial than using one or no layers. Moreover, all of our EVaDE layers are capable of learning the identity function (Theorem 1). Thus, given enough data and that the optimization process does not get stuck in a local optimum, it is possible that the agents learn ‘not’ to use a certain layer for exploration if it is not useful.
>
> &nbsp;
> - Why approximating a posterior for the reward function part only? Wouldn't it be also possible to use it for the dynamics?\
>  In our initial experiments, we observed that the agents learnt better policies when samples from the posterior were drawn by perturbing only the reward function instead of both the transition and reward functions. It is much easier to use prior knowledge on reward functions. Indiscriminate exploration can hurt performance and we do not have sufficiently good knowledge on how to construct good variational distributions on the transitions, whereas we are able to do so for the reward function. Thus, the decision to carry out the whole set of experiments, with the perturbations applied to the reward model alone, was made empirically. Furthermore, as the generated trajectory depends on the transition model, we feel that the imperfections of the transition model, which compound over the length of a generated episode, could be further exaggerated when perturbations are applied.
>
> &nbsp;
> - Why the horizon length tradeoff influences the number of iterations.\
> Our agents work in a limited data regime, where it can have only 100K interactions in total with the real environment. Additionally, we collect an equal number of interactions in every iteration. Thus, a larger horizon length in every iteration would correspond to less iterations in EVaDE-SimPLe, as the total number of interactions is fixed.
>
> &nbsp;
> - "we train the environment model for 45K steps in the first iteration and 15K steps in all subsequent iterations": What does this mean?\
> The environment model is trained with the help of supervised learning. In the first iteration, we use the dataset containing 6400 interactions to train the environment model with 45K batches of 4 interactions drawn from this dataset. In all other iterations, we train the model using 15K batches, where the training batches are drawn from the dataset of all real environment interactions collected until that iteration.

---

> > ### Author Response · Authors · 2021-11-14
> > **Response to Reviewer XiQT (Part 2 of 2)**
> >
> > - Why is "z" changing?\
> > We try to follow the training procedure for SimPLe(30) as closely as possible[3]. However, since we use 30 iterations instead of 15, we replicate the training schedule from iterations 1 to 14 twice and use z=3 in the last iteration as is done in SimPLe.
> > We train EVaDE-SimPLe using the same training procedure above so that we can attribute the performance improvements solely to the noisy layers and the process of acquiring posterior samples.
> >
> > We also thank the reviewer for the other suggestions with regards to clarity.
> >
> > [1] Lu, Xiuyuan, and Benjamin Van Roy. "Ensemble sampling." arXiv preprint arXiv:1705.07347 (2017).
> >
> > [2] Zhang, Ruiyi, et al. "Scalable thompson sampling via optimal transport." arXiv preprint arXiv:1902.07239 (2019).
> >
> > [3] Kaiser, Łukasz, et al. "Model Based Reinforcement Learning for Atari." International Conference on Learning Representations. 2019.

---

> > > ### Comment · Reviewer_XiQT · 2021-12-01
> > > **Thank you for the reply**
> > >
> > > I want to thank the authors for their reply to my questions.

---

### Official Review · Reviewer_gXzj · 2021-10-30

**Correctness:** 4
**Technical Novelty And Significance:** 2
**Empirical Novelty And Significance:** 2
**Recommendation:** 5
**Confidence:** 3

**Main Review:**

This paper proposed some new ideas on neural network architecture construction built on the algorithm of PSRL. But overall I feel this paper does not tackle the key problem of reinforcement learning. Both the exploration strategy and planning algorithm are the same as previous works. The novel design of NN architecture is a bit too specific and lacks of principles. As a research paper, I feel it should provide more principle algorithm design ideas rather than adaptation on one or two specific domains. This may lose the generality.

The approach described in Section 3 is a bit heuristics and not very related to RL. Why posterior sampling is done by multiplying
each parameter of these EVaDE layers by a perturbation drawn from a Gaussian distribution? What is the posterior here? In section 3.2, dropout is introduced. It's a bit hard to understand how you approximate the posterior of the model in the end. I hope the authors could organize the section in a more mathematic way.

I respectively disagree on "an interesting aspect of designing for exploration is that the variational distributions can be helpful
even if they are not designed to approximate the posterior well, as long as they assist in perturbing the policy out of local optimums." This is still a very heuristic argument and lacks of theoretical support. PSRL has a strong theoretical guarantee and people then can spend effort on better approximation for posterior distribution. I feel this is a more principle way for exploration.

"it is easier to incorporate inductive biases derived from the domain knowledge of the task for learning the model, as the biases can be directly built into the transition and reward functions." I do not quite understand how you can do this and how it is supported. Especially "easier" with respect to what? Has this point been discussed anywhere else in the paper?

"Model-free agents explore the space of policies" What do you mean by that? How about value-based methods? I hope the authors could use accurate and well-supported arguments.

"object-based" and "event" should be clearly defined in the beginning.

I feel it should be necessary to have a full algorithmic box for Section 3.5 for self-completeness. How do you interact with the environment and how do you do the planning?

**Summary Of The Paper:**

This paper studied exploration problems in model-based reinforcement learning. Some new neural network architecture are provided to incorporate object-based domains.

**Summary Of The Review:**

This paper provides a novel neural network architecture design for model-based RL exploration. Overall, I feel it lacks principle explanation and is quite specific to a small set of domains.

---

> ### Author Response · Authors · 2021-11-14
> **Response to Reviewer gXzj**
>
> We thank the reviewer for the detailed and informative review and pertinent suggestions. We address the questions and concerns raised below.
>
>
> - The work is not very novel and lacks generality.\
> To the best of our knowledge, this is the first work that has successfully used dropout induced variational distributions for Thompson sampling in  MBRL methods.  We agree with the reviewer that principled methods are desirable, however, we believe that empirical work can be important for progress in an area. Our method provides a practical way to add randomness to the SimPLe world model with significant performance improvements.
>  Moreover, EVaDE-SimPLe is not designed to be specific to Atari games only and can be used in any domain that deals with objects and their interactions. As such domains where objects interact with each other are common, we expect the method to be widely applicable. We use Atari only as a testbed, as it is a popular domain to test RL methods. The authors of SimPLe also claim that it can handle other visual prediction tasks [1]. By extension, we believe that EVaDE-SimPLe can do it as well, as it is a method that augments SimPLe with approximate Thompson Sampling. Thus, we believe that EVaDE is a fairly general method for deep neural network based models.
>
>  &nbsp;
> - The paper just relies on heuristics, does not provide a principled algorithm and lacks theoretical support.\
> The use of variational methods to approximate posterior distributions is a principled method for approximation and is relied upon by several methods (Some examples include [3,4,5]). In addition, dropout has been shown to induce variational distributions that approximate posterior distributions. [2] links dropout and Gaussian processes, to show that adding dropout while training a neural network model amounts to approximating Bayesian inference in deep Gaussian processes. These Bayesian properties of dropout can be helpful in  constructing variational distributions that approximate Bayesian posteriors [6] . We utilize these properties of dropout to induce variational distributions that approximate the posterior of reward models.  We also provide some background on the related technical details in Appendix C.
>
>  &nbsp;
> - Why is it easier to incorporate domain knowledge in environment models than in policy models?\
>  Reward functions are often sparse and non-zero or non-constant, for example only when objects interact -- these intuitions can be  directly incorporated in model based methods, whereas to to incorporate them in model-free methods may require non-trivial look-ahead to compute the effects of acting optimally until the objects interact.
>
>  &nbsp;
> - "Model-free agents explore the space of policies" What do you mean by that? How about value-based methods? I hope the authors could use accurate and well-supported arguments. \
> We agree with the reviewer on this point and will modify this statement to incorporate both values and policies.
>
>  &nbsp;
> - Full algorithmic box for Section 3.5 \
> Unfortunately, due to the page limit of 9 pages for the main submission, we could not provide an algorithmic box for Section 3.5. We have, however, added it in Appendix A.
>
> We also thank the reviewer for the other suggestions regarding clarity.
>
>
>
> [1] Kaiser, Łukasz, et al. "Model Based Reinforcement Learning for Atari." International Conference on Learning Representations. 2019.
>
> [2] Gal, Yarin, and Zoubin Ghahramani. "Dropout as a bayesian approximation: Representing model uncertainty in deep learning." international conference on machine learning. PMLR, 2016.
>
> [3] Aravindan, Siddharth, and Wee Sun Lee. "State-Aware Variational Thompson Sampling for Deep Q-Networks." arXiv preprint arXiv:2102.03719 (2021).
>
> [4] Lamprier, Sylvain, Thibault Gisselbrecht, and Patrick Gallinari. "Variational Thompson sampling for relational recurrent bandits." Joint European Conference on Machine Learning and Knowledge Discovery in Databases. Springer, Cham, 2017.
>
> [5] Tang, Yunhao, and Alp Kucukelbir. "Variational Deep Q Network." (2017).
>
> [6] Kingma, Durk P., Tim Salimans, and Max Welling. "Variational dropout and the local reparameterization trick." Advances in neural information processing systems 28 (2015): 2575-2583.

---

### Official Review · Reviewer_trzP · 2021-11-02

**Correctness:** 3
**Technical Novelty And Significance:** 2
**Empirical Novelty And Significance:** Not applicable
**Recommendation:** 5
**Confidence:** 3

**Main Review:**

This paper proposes a straightforward and easy-to-implement modification to the SimPLe algorithm (Kaiser et al., 2020) - modify three layers in the transition and reward model they are using. This simplicity is good, but it requires much more rigorous experimentation since users of such simple modifications would want to know why and when these modifications work or don’t work. Otherwise, there are infinite tweaks one can do to an architecture - but which one actually helps? My main concern is that there are only 3 independent runs from which it is difficult to judge whether EVaDE is really better than SimPLe. In the ablations, it even seems that sometimes, using just one of the layers outperforms using all three.


**Summary Of The Paper:**

This paper improves a model-based reinforcement learning method by Kaiser et al. (2020) by introducing three different types of neural network layers to the transition and reward model. These modifications are motivated by intuitions about what would be beneficial to model for object-based environments (in Atari games). There are experiments on 12 games in which the proposed method performs the best most of the time. Ablations show that using any one of the three proposed layers is also beneficial.


**Summary Of The Review:**

A simple modification to an existing architecture is nice, but it’s unclear from the experiments that this particular modification is really helping, due to the small amount of seeds.

---

> ### Author Response · Authors · 2021-11-14
> **Response to Reviewer trzP**
>
> We thank the reviewer for the comments. We address some of the points raised by the reviewer below:
>
> - Why only 3 independent runs? \
> We conducted 3 independent training runs for each method, as most methods that work in the 100k setting work with only 3 or 5 training runs [2,3,4]. With limited resources at our behest, we aimed to test the generality in a broader set of Atari games with three seeds. We also provide the results of individual runs in Appendix E.2 . Should we have decided to increase the number of seeds, we would have had to test the suite on a smaller set of games, which would have been less convincing. We provide details about the amount of compute used in Appendix E.5.
> However, given the above limitation, we are still confident that EVaDE-SimPLe has considerable performance improvements over SimPLe(30). \
> To reinforce this statement, we report 2 additional statistics that confirm the same :
>     - We conducted a paired t-test on the mean human normalized scores achieved by EVaDE-SimPLe and SimPLe(30) on each of the 12 games. We got a single-tailed p-value of 0.018 from this test, which confirms that the mean scores of EVaDE-SimPLe and SimPLe(30) are different with very high probability.
>     - As suggested in a public comment, we also compute the Inter-Quartile Mean [1] of the EVaDE-SimPLe and SimPLe(30) training runs. EVaDE-SimPLe achieves an IQM score of 0.42, while SimPLe(30) achieves an IQM score of 0.23.
>
>    While 3 runs may not be statistically significant for a single game, the paired t-test results  show that the improvements of EVaDE-SimPLe is statistically significant as an algorithm when applied to multiple games. Moreover, EVaDE-SimPLe also achieves a significantly higher score as compared to SimPLe(30) on IQM, a metric robust to outliers.
>
>  &nbsp;
> - How do the EVaDE layers work? \
> We try to provide an intuitive justification for why the EVaDE layers should work in Section 3. Additionally, in Appendix E.3, we provide several visualizations that help us visualize the functionality of these layers.
>
> &nbsp;
> - Why does using just one EVaDE layer outperform all three for some games? \
> It is possible that some layers may be less effective in some games, leading to the agent wasting its interaction budget exploring areas suggested by one of the layers that is less effective for that particular game. However, in general, a combination of all three EVaDE layers is more beneficial than using a single layer, as we see that it achieves a better HNS on average. Moreover, all of our EVaDE layers are capable of learning the identity function (Theorem 1). Thus, given enough data and that the optimization process does not get stuck in a local optimum, the agents could learn ‘not’ to use particular layers for exploration if they are not helpful.
>
> &nbsp;
> [1] Agarwal, R., Schwarzer, M., Castro, P.S., Courville, A. and Bellemare, M.G., 2021. Deep reinforcement learning at the edge of the statistical precipice. In NeurIPS.
>
> [2] Kaiser, Łukasz, et al. "Model Based Reinforcement Learning for Atari." International Conference on Learning Representations. 2019.
>
> [3] Steven Hansen, Will Dabney, Andre Barreto, David Warde-Farley, Tom Van de Wiele, and Volodymyr Mnih. Fast task inference with variational intrinsic successor features. In International Conference on Learning Representations, 2020.
>
> [4] Hado van Hasselt, Matteo Hessel, and John Aslanides. When to use parametric models in reinforcement learning? NeurIPS, 2019.

---

> > ### Comment · Reviewer_trzP · 2021-12-01
> > **Thank you**
> >
> > I have increased my score in light of the author responses as I was not aware that it is standard in RL to only evaluate very few (3-5) runs. I am hesitant to increase to a higher score as I still feel that the architectural changes to the neural network, which form a large part of the conceptual contribution, seem very specific.

---

> > > ### Author Response · Authors · 2021-12-02
> > > **Authors' Response**
> > >
> > > We thank reviewer trzP for their reply and for increasing the score.
> > >
> > > We would like to address the concern raised and clarify that EVaDE-SimPLe is not designed to be specific to Atari games only and can be used in any domain that deals with objects and interactions. Moreover, as domains with objects and interactions between objects are common, we expect the method to be widely applicable. In fact, we use Atari only as a testbed, as it is a popular domain to test RL methods. The authors of SimPLe also claim that it can handle other visual prediction tasks. By extension, we believe that EVaDE-SimPLe can do it as well, as it is a method that augments SimPLe with approximate Thompson Sampling. Thus, we believe that EVaDE is a fairly general method for deep neural network based models.

---

### Official Review · Reviewer_9oaA · 2021-11-05

**Correctness:** 4
**Technical Novelty And Significance:** 3
**Empirical Novelty And Significance:** 3
**Recommendation:** 8
**Confidence:** 3

**Main Review:**

One key challenge in RL is exploration. Posterior sampling or Thompson sampling has shown good theoretical properties for exploration in RL, and has good performance in certain RL exploration problems. This paper applies the posterior sampling idea to the model-based deep RL setting.

In order to apply posterior sampling to model-based deep RL, one need to maintain an approximate posterior distribution of the learned deep neural net model. The paper proposed a variational approach, called Event based Variational Distributions for Exploration (EVaDE), which consists of three types of Gaussian dropout convolutional layers. The three types of layers attempt to capture three aspects in a game: interaction between objects, random event emphasis, and random object translation. These dropout layers induced variational distributions are used to approximate the posterior distribution which is then used in posterior sampling for exploration.

The proposed dropout layers for variational distributions are appealing, and the ideas to use three types of layers to capture three kinds of effects is very interesting. But little is discussed in the paper on why to consider these three effects. There are some interesting visualizations for the layers in the appendix, but there is no quantitative analysis on whether each layer type indeed performs what it is designed to do. The ablation study is good to suggest the benefits of adding each layer type, but little is shown on what is the effect of each type of the layers. Does a certain type of layer help exploring a specific kind of hard-explored situation among the games?

In the experiments, the paper equips the EVaDE layers to an existing model-based RL method SimPLe and compare EVaDE-SimPLe with four baselines in 12 Atari games. EVaDE-SimPLe achieves good performance compared with the baselines, but It would be better if the paper provides more discussions on why the proposed method performs better in some games but not in some others. Does some games more difficult for exploration? Does the noisy layers hinder the agent's performance in some situations? In particular, in JamesBond, EVaDE-SimPLe performs worse than SimPLe and the ablation study seems to suggest no benefits from any layer in this game. Can EVaDE-SimPLe achieve the same performance if trained longer, or the noisy layers prevent the agent to achieve the best performance when exploration is less important?

**Summary Of The Paper:**

The paper proposes a method to equip model-based deep reinforcement learning with posterior sampling for exploration where posterior sampling is approximated using a variational distribution approach.

**Summary Of The Review:**

Pros
* Good idea to apply posterior sampling to model-based deep RL.
* Novel Gaussian dropout layers capturing objects for important events.
* Good experimental performance in 12 Atari games.

Cons
* Not enough explanations and discussions for the three proposed layers.
* No discussion on why EVaDE-SimPLe performs worse than SimPLe in some games.

---

> ### Author Response · Authors · 2021-11-14
> **Response to Reviewer 9oaA**
>
> We thank the reviewer for the detailed and informative review, pertinent suggestions and positive comments. We address some of the points raised by the reviewer below.
>
> - Quantifying how EVaDE layers work. \
>  We agree with the reviewer that quantifying why and how the noisy layers function could help determine their applicability for different tasks. However, we would like to emphasize that since these layers are inserted in between any two existing hidden layers of the reward model, it could be challenging to interpret the abstract representations of the objects/events captured by the intermediate outputs and thus comment on the above. As an alternative, we provide some visualizations that help us understand the function of these layers. We also perform ablation studies to compare the policies learnt when each of these layers was added individually to show that these structural biases help learn better policies.
>
> &nbsp;
> - Why do EVaDE layers help in some games more than others? \
>  It is possible that some layers are less effective in some games, leading to the agent wasting its interaction budget exploring areas suggested by one of the less effective layers for that particular game. As such, in the two games, it is possible that the layers explore excessively, leading to lower scores. However, on average, using all three EVaDE layers seems more beneficial than using one or no layers. This could also be the reason why we see the delay in the learning curves of some EVaDE-SimPLe agents as detailed in the ablation studies.
>  Moreover, all of our EVaDE layers are capable of learning the identity function (Theorem 1). Thus, given enough data and that the optimization process does not get stuck in a local optimum, the agents may learn ‘not’ to use a particular layer for exploration if it is not helpful.

---

### Public Comment · ~Rishabh_Agarwal2 · 2021-11-09
**Large statistical uncertainty in Atari 100k results**

Hi authors,

The case study in [1] on Atari 100k shows that results on this benchmark has significant variance in results with overlapping standard deviations for various games and even changes in ranking comparisons across methods when using 100 seeds. See [this figure](https://pbs.twimg.com/media/E-IkSDdXMAUV1dx?format=jpg&name=large) for an example. However, the current results for Atari 100k ignore the statistical uncertainty in aggregate human normalized scores and report the per-game mean scores for comparison in the main Table.

For reliable evaluation, I'd recommend the authors to follow the reliable evaluation protocols suggested in [1] when using only a few seeds such as reporting score distributions and aggregate performance metrics like IQM with confidence intervals (CIs). Similarly, the number of games with best ranking can be replaced by the probability of being the best ranked -- the probability of improvement metric with CIs can be a simpler alternative to show the same thing. You can easily do so using the library at https://github.com/google-research/rliable or the corresponding [colab](https://bit.ly/statistical_precipice_colab).

Minor point: CURL achieves a lower score than reported as CURL's reported scores were based on maximum performance during training rather than end performance results. Please find the results for 100 runs for CURL/OTRainbow here: https://console.cloud.google.com/storage/browser/rl-benchmark-data/atari_100k.

[1] Agarwal, R., Schwarzer, M., Castro, P.S., Courville, A. and Bellemare, M.G., 2021. Deep reinforcement learning at the edge of the statistical precipice. In NeurIPS.

---

### Author Response · Authors · 2021-11-20
**Paper Revision**

We have modified our manuscript with the following changes as per the reviewers' comments.

- Moved the description of what "object-based" and "events" mean from Section 3 to the Introduction (Page 2, paragraph2).
- Modified the statement "Model-free agents explore the space of policies" to "Model-free agents explore the space of policies and value functions".
-Corrected Table 1 title.
-Added that the agents are initialized with a random policy before collecting the first 6400 interactions.

---

### Author Response · Authors · 2021-11-23
**Thank You**

We thank all the reviewers for their informative comments and constructive suggestions.
We hope that our modifications to the manuscript (summarized in the comment below) and the clarifications provided as replies to the reviews have assuaged most concerns/queries raised by the reviewers and welcome any feedback regarding the same. We will be happy to address any other queries that arise during the rest of the review period.
We again thank the reviewer committee for their time and effort.

---

### Author Response · Authors · 2021-11-28
**Looking forward to feedback on review responses**

We hope that our updated manuscript and detailed responses have addressed your comments, queries and suggestions. As there are less than two days left in the discussion period, could the reviewers please let us know if there are any remaining concerns or questions that we can address?

We thank the review committee again for their time and effort!

---

### Decision · Program_Chairs · 2022-01-20

**Decision:**

Reject

**Comment:**

This paper proposes to implement posterior sampling for reinforcement learning for MBRL using three types of noisy convolutional layers inspired by object- and event-based domain knowledge. These layers are used to augment the SimPLe agent (Kaiser et al, 2020), resulting in the EVaDE-SimPLe agent, and experiments demonstrate that the EVaDE-SimPLe outperforms SimPLe on average across twelve Atari games.

The reviewers' opinions on the paper were mixed. The reviews highlighted several strengths of the paper: that using posterior sampling for exploration in MBRL is well-motivated (Reviewers 9oaA, XiQT) and that the simplicity of the proposed layers is appealing (Reviewers trzPm, XiQT). However, the reviewers also generally felt that the proposed method was overly specific to a particular domain (Reviewer gXzj, XiQT) and that there was not enough analysis demonstrating *why* the proposed layers work, in which cases they would not work, or why these modifications might be better than other similar modifications (Reviewers 9oaA, trzP, gXzj). Initially there were also some concerns raised by Reviewer trzP about the validity of the evaluation due to the number of seeds, though these concerns were addressed by the authors during the rebuttal.

I agree with the reviewers that the approach is interesting and that getting posterior sampling to work well in MBRL is an important problem. But I also find myself agreeing that the present approach is not analyzed in sufficient depth (the results are overly focused on just overall performance, rather than analyzing behaviors exhibited by the agents) and that it is unclear how well it would work in other domains (e.g. 3D settings). I therefore feel this work is not quite ready to be presented at ICLR, and recommend rejection.